# Cancer-Associated Fibroblasts in Pancreatic Ductal Adenocarcinoma or a Metaphor for Heterogeneity: From Single-Cell Analysis to Whole-Body Imaging

**DOI:** 10.3390/biomedicines12030591

**Published:** 2024-03-06

**Authors:** Rita Saúde-Conde, Ayça Arçay Öztürk, Kosta Stosic, Oier Azurmendi Senar, Julie Navez, Christelle Bouchart, Tatjana Arsenijevic, Patrick Flamen, Jean-Luc Van Laethem

**Affiliations:** 1Digestive Oncology Department, Hôpitaux Universitaires de Bruxelles (HUB), Université Libre de Bruxelles (ULB), 1070 Brussels, Belgium; 2Laboratory of Experimental Gastroenterology, Université Libre de Bruxelles, 1070 Brussels, Belgium; 3Nuclear Medicine Department, Hôpitaux Universitaires de Bruxelles (HUB), Institut Jules Bordet, Université Libre de Bruxelles (ULB), 1070 Brussels, Belgium; patrick.flamen@hubruxelles.be; 4Department of Abdominal Surgery and Transplantation, Hôpitaux Universitaires de Bruxelles (HUB), Hopital Erasme, Université Libre de Bruxelles, 1070 Brussels, Belgium; 5Department of Radiation Oncology, Hôpitaux Universitaires de Bruxelles (HUB), Institut Jules Bordet, Université Libre de Bruxelles (ULB), 1070 Brussels, Belgium

**Keywords:** cancer-associated fibroblast, Gallium-68 fibroblast activation protein inhibitor positron emission tomography, fibroblast activation protein, pancreatic cancer, heterogeneity

## Abstract

Pancreatic ductal adenocarcinoma (PDAC) represents a formidable challenge due to its aggressive nature and poor prognosis. The tumor microenvironment (TME) in PDAC, characterized by intense stromal desmoplastic reactions and a dominant presence of cancer-associated fibroblasts (CAFs), significantly contributes to therapeutic resistance. However, within the heterogeneous CAF population, fibroblast activation protein (FAP) emerges as a promising target for Gallium-68 FAP inhibitor positron emission tomography (Ga68FAPI-PET) imaging. Notably, 68Ga-FAPI-PET demonstrates promising diagnostic sensitivity and specificity, especially in conjunction with low tracer uptake in non-tumoral tissues. Moreover, it provides valuable insights into tumor–stroma interactions, a critical aspect of PDAC tumorigenesis not adequately visualized through conventional methods. The clinical implications of this innovative imaging modality extend to its potential to reshape treatment strategies by offering a deeper understanding of the dynamic TME. However, while the potential of 68Ga-FAPI-PET is evident, ongoing correlative studies are essential to elucidate the full spectrum of CAF heterogeneity and to validate its impact on PDAC management. This article provides a comprehensive review of CAF heterogeneity in PDAC and explores the potential impact of 68Ga-FAPI-PET on disease management.

## 1. Introduction

Pancreatic ductal adenocarcinoma (PDAC) is an aggressive malignancy associated with high metastatic risk or locally advanced disease precluding surgery, leading to a poor prognosis [1]. PDAC ranks fourth in the cancer mortality classification, and according to some estimations, it may reach the second position by 2030, especially in the western world [2].

The tumor microenvironment (TME) of PDAC stands out as notably distinct, intricate, and highly dynamic compared to other solid tumors. This complexity arises from extensive interactions among various TME components, providing numerous avenues for resistance and heightened tumor aggressiveness. The TME in PDAC plays a significant role in the pronounced resistance observed against conventional and immune therapies [3,4]. PDAC tumors are characterized by prominent stromal desmoplastic reactions surrounding cancer cells, resulting in an acidified and hypoxic microenvironment with elevated interstitial pressure. This creates a drug-resistant sanctuary and poses a physical barrier to immune cells recruitment [4,5,6,7]. Central to the desmoplastic reaction are cancer-associated fibroblasts (CAFs), representing a significant component of the TME, accounting for up to 90% of the tumor tissue in PDAC. 

The CAF population exhibits both inter- and intra-tumoral heterogeneity in pathways, marker genes, functions, and spatial distribution [8,9,10,11,12]. Despite the potential for both pro- and anti-tumorigenic functions within the diverse CAF population, targeting CAFs remains an attractive strategy due to their substantial presence in the TME and their genetic stability. This makes them prime targets, being less likely to develop resistant phenotypes compared to tumor cells [4,5,7,13,14,15]. 

Fibroblast activation protein (FAP), a type-II transmembrane serine protease, is highly expressed in CAFs [16]. This extensive expression renders FAP an attractive target for radionuclide imaging with positron emission tomography/computed tomography (PET/CT) [6,7,17,18]. FAP-targeted PET imaging, characterized by diagnostic sensitivity and specificity, has shown promise, with low uptake of FAP-targeted tracers in non-tumoral tissues and high uptake in a wide range of cancers, PDAC among them [19]. Given the significant presence of CAFs in PDAC, intensive uptake of Gallium-68 FAP inhibitor (Ga68-FAPI) is expected, as demonstrated in previous studies showing a promising clinical value of Ga68-FAPI PET/CT for pancreatic cancer diagnosis [20,21]. 

The application of Ga68-FAPI-PET offers unique insight into tumor–stroma interactions, which is essential for understanding PDAC tumorigenesis and not adequately observed through conventional imaging. Thus, the use of Ga68-FAPI-PET in PDAC holds special interest, potentially leading to significant changes in treatment decisions, and it may be a key innovative player in improving PDAC management. 

This article will first review the available evidence regarding CAF heterogeneity and plasticity in PDAC. Second, it will discuss the potential impact of Ga68-FAPI-PET in PDAC on disease understanding and management. Finally, it will examine the promises and challenges of FAP-targeted radioligand therapy in PDAC, particularly in the neoadjuvant context, emphasizing ongoing clinical trials.

## 2. Overview of Cancer-Associated Fibroblasts (CAFs) in PDAC: Heterogeneity and Plasticity

### 2.1. TME and CAFs

First identified in the early 20th century, acknowledgments that the TME is as important as the tumor cell itself have been increasing [22,23,24]. 

The TME can be defined as the surrounding ecosystem where tumor cells reside, and it is composed of a cellular and an acellular component [24,25,26]. The acellular component includes the extracellular matrix (ECM), extracellular vesicles (exosomes, apoptotic bodies), and extracellular molecules (cytokines growth factors). The cellular component consists of non-neoplastic cells like immune cells (macrophages, polymorphonuclear cells, natural killer cells, dendritic cells, T and B lymphocytes, etc.) and non-immune cells (endothelial cells, neuroendocrine cells, adipocytes, mesenchymal stem cells, and cancer-associated fibroblasts) [23,24].

Fibroblasts in normal tissues are generally spindle-shaped single cells without the association of a basement membrane but embedded within the fibrillar ECM of the interstitium, present in the interstitial space or near a capillary [27]. They play a major role in wound healing, where they transform from a quiescent form to an activated contractile form (myofibroblasts) to maintain tissue homeostasis [27,28]. Fibroblasts have been identified in various tumor types and have hence generally been termed CAFs [29]. CAFs are perpetually activated fibroblasts with a high capacity to synthesize ECM components in tumors, a phenomenon called stromal desmoplasia [27,30,31]. However, CAFs are actually a highly heterogeneous and plastic population within the TME.

### 2.2. CAFs and Cellular Origin Heterogeneity

CAFs are defined by the exclusion of cells with a notable lack of lineage markers for epithelial cells, endothelial cells, and leukocytes and without the mutations typically found in cancer cells [31]. Due to the lack of exclusive cell markers, their cellular origin remains imprecise. In PDAC, CAFs are thought to originate from multiple origins such as resident fibroblasts like pancreatic stellate cells (PSC), or non–fibroblast lineage cells such as epithelial cells, endothelial cells, adipocytes, and pericytes [32]. 

Resident fibroblasts, especially PSC, have the ability to transdifferentiate from a “quiescent” retinoid/lipid storing phenotype in the normal pancreas to an “activated” α-smooth muscle-actin-producing myofibroblastic phenotype through tumor-derived stimuli such as cytokines (interleukin(IL)-1, IL-6, and IL-8 and tumor necrosis factor (TNF)-α), growth factors (platelet-derived growth factor (PDGF) and tumor growth factor (TGF)-β), and reactive oxygen species [33]. Activated PSCs can, in turn, produce autocrine factors such as PDGF, TGF-β, and cytokines, which may contribute to a looping mechanism promoting a desmoplastic reaction [34]. Interestingly, conflicting results arise from an in vivo study, where PSCs gave rise to a numerically minor subset of PDAC CAFs, suggesting a greater variability in CAF precursors [35]. Accordingly, normal resident fibroblasts, which reside around the tumor cells, have been found to be activated via the tumor cell-derived signaling pathway and give rise to a subset of CAFs in PDAC [10]. In another recent study, genetically engineered models using a dual-recombinase approach were used to follow two normal fibroblast populations marked by the expression of Gli1+ or Hoxb6+, two mesenchymal markers that have not been associated with stellate cells. Although Gli1+ and Hoxb6+ fibroblasts are equally present in the healthy pancreas, they appear to mark a distinct fibroblasts population with minimal overlap; that is, Gli1+ fibroblasts expand dramatically, whereas Hoxb6+ fibroblasts do not appear to give rise to CAFs [36]. These findings indicate that heterogeneity might be predetermined based on the type of progenitor cell from which CAFs arise. 

In addition, CAFs may arise from multiple non-fibroblast lineage cells such as epithelial and endothelial cells, through the epithelial-to-mesenchymal transition (EMT) or endothelial-to-mesenchymal transition (endMT), respectively, or also less commonly, from adipocytes, pericytes, smooth muscle cells [27,37,38,39,40]. Accordingly, a recent study indicated that antigen-presenting CAFs (apCAFs) may arise from mesothelial cells through the IL-1- and TGFβ-induced down-regulation of mesothelial features and upregulation of fibroblastic ones [41].

Although predominantly observed to be of local origin, the potential origin of bone marrow has been studied in several tumor types, such as PDAC [42]. Mesenchymal stem cells from the bone marrow can differentiate into a subpopulation of CAFs under TGF-β, WNT, and IL-6/STAT3 signaling [43]. Similarly, bone marrow macrophages/monocytes can convert into CAFs [44]. The multiple cells of origin from which CAFs are suggested to derive are presented in Figure 1.

### 2.3. CAFs Phenotypical Heterogeneity

Studies of human cancers and mouse models using immunostaining, in situ hybridization, flow cytometry, fluorescence-activated cell sorting, and mRNA microarrays were crucial to establish the existence of CAFs subpopulation. More recently, the advent of single-cell RNA-sequencing (scRNAseq) technologies has further enabled an improved characterization of the complexity and heterogeneity of CAF subpopulations [45].

Öhlund’s ancillary study used a three-dimensional in vitro co-culture system, consisting of PSCs and KPC (KrasLSL-G12D; Trp53 LSL-R172H/+; Pdx1Cre/+) mouse-derived PDAC organoids, and distinguished two distinct CAF subpopulations: myofibroblastic CAFs (myCAFs) and inflammatory CAFs (iCAFs) [12]. Later, Elyada et al. utilized a droplet-based scRNAseq approach to confirm the existence of both CAF subpopulations but also to find a third CAF subtype: apCAF [9]. TGF-β, by downregulating IL1 receptor (IL1R1) expression, promotes fibroblast transformation to myCAFs and inhibits transformation to iCAFs [46].

#### 2.3.1. myCAFs

Before their characterization by RNAseq, CAFs were already recognized as being associated with myofibroblasts owing to their activated state wherein they develop specialized contractile traits akin to those observed in fibroblasts during wound healing processes [28,45]. Upon acute injury, resident fibroblasts are activated through TGF-β signaling and evolve into myofibroblasts, expressing high levels of α-smooth muscle actin (α-SMA) [47].

Öhlund et al. analyzed the human PDAC fibroblast via immunofluorescence to identify PDAC fibroblasts utilizing FAP and α-SMA, revealing that that while all fibroblasts exhibited FAP, a larger proportion displayed minimal levels of α-SMA, with a minority exhibiting substantial levels of α-SMA, thus confirming the hypothesis of heterogeneity [12]. Secondly, they found that the high-level α-SMA CAFs expressed low-levels of IL6 and were only activated when tumor cells came into direct contact with PSCs [12]. Those CAFs, which represent around 50% of all CAFs, were named myCAFs. Additionally, and in line with two other studies, the transcriptome of myCAFs was compared to quiescent PSCs in vitro and showed an upregulation of ACTA2 (α-SMA), CCN2 (CTGF), and COL1A1 [12,48,49]. ScRNAseq revealed a cluster of novel marker genes encoding contractile proteins, such as transgelin (TAGLN), myosin regulatory light chain 9 (MYL9), tropomyosin 1 and 2 (TPM1 and 2), and periostin (POSTN) [9]. Additionally, activated proteins in myCAFs encompassed TGF-β1, SMAD family member 2 (SMAD2), and Twist family BHLH transcription factor 1 (TWIST1) [9]. Biffi et al. showed that TGFβ suppresses the IL-1 receptor 1 (IL1R1), which activates the SMAD2/3 pathway and promotes differentiation into myofibroblasts. In summary, myCAFs are induced by the TGF-β/SMAD2/3 pathway [46].

#### 2.3.2. iCAFs

As previously mentioned, Öhlund et al. also found a second CAF subtype, named iCAFs, which presented low levels of α-SMA and high levels of IL6, with a loss of myofibroblastic features. Elyada and colleagues confirmed their existence and found new upregulated marker genes, such as IL6, IL8, IL11, leukemia inhibitory factor (LIF), and chemokine (C-X-C motif) ligand 1 (CXCL1) and 2 (CXCL12) [9,12]. Additionally, iCAFs specifically expressed hyaluronic acid synthase 1 (HAS1) and HAS2, two enzymes playing a major role in the synthesis of hyaluronic acid, which is paramount in drug resistance [9]. 

Biffi et al. showed that IL1, secreted by tumor cells, acts through nuclear factor kappa-B (NF-κB) and IL6, induces expression of the LIF, and activates downstream Janus kinase/signal transducer and activator of transcription (JAK/STAT) to generate iCAFs [46]. Another study showed that LIF, secreted by PSCs, was associated with tumor progression in PDAC [50]. In summary, the IL1/LIF/JAK–STAT3 pathway could induce the iCAFs.

#### 2.3.3. apCAFs

Antigen-presenting cells (APCs), typically dendritic cells, macrophages, and B cells usually express major histocompatibility complex (MHC) class II family genes and have the ability to activate T cells. Through scRNAseq, RNA in situ hybridization, immunohistochemistry (IHC), and imaging mass cytometry, Elyada et al. identified apCAFs [9] CAFs that exhibited distinctive genes expression, encompassing H2-Aa, H2-Ab1, and CD74 (encoding chains of MHC II), serum amyloid A3 (Saa3), and secretory leukocyte peptidase inhibitor (SLPI) [10]. Hosein et al. demonstrated that apCAFs were capable of antigen processing and presentation via the MHC-II pathway, along with possession complement activation functions [49]. However, unlike professional APCs, apCAFs poorly expressed classic co-stimulatory molecules such as CD40, CD80, or CD86, indicating they act differently from professional APCs [49]. Hence, the authors hypothesize that the MHC II expressed by apCAFs might function as a decoy receptor, disrupting the interaction with CD4+ T cells and thereby inhibiting their clonal proliferation, potentially including T-cell anergy or promoting differentiation into Tregs, consequently fostering an immunosuppressive TME [29]. Moreover, apCAFs demonstrated a higher activity for STAT1, typically associated with IFNγ signalling in vivo, and displayed an anti-oxidant response [9]. 

With single-nucleus RNA sequencing (snRNAeq), Regev et al. profiled 224,988 nuclei across 43 PDAC specimens (18 untreated and 25 treated) and used a refined molecular stratification to identify four distinct programs of CAFs in PDAC [9,51]. Among those programs historically identified by Elyada et al., the ACTA2-enriched myofibroblastic progenitor program overlapped with Elyada’s myCAF signature but is distinguished by an enrichment of genes involved in embryonic mesodermal development and Wnt signaling. The neurotropic, immunomodulatory, and adhesive programs overlap with the single-cell iCAF signature, indicating potential different iCAF subsets, albeit with no significant overlap with the myCAF signature. In addition, the CAF programs exhibit non-specific overlap with various cross-tissue fibroblast signatures [51]. Additional investigations are warranted to delve deeper into the diverse molecular characteristics of CAFs in PDAC. Further studies can provide a more comprehensive understanding of the intricate interplay between different CAF subsets and their potential roles within the TME.

#### 2.3.4. Other CAF Subtypes

Recently, a novel CAF subtype named metabolic CAF (meCAF), was discovered in PDAC human tissue with loose ECM (low desmoplasia), where meCAFs are suggested to be the most abundant CAF subtype. They are characterized by the expression of PLA2G2A, while their differentiation is potentially CREB3L1-dependent, and show a highly glycolytic activity [52].

Another recently described CAF subtype is the Meflin+ CAF, which displays low α-SMA expression [53]. Meflin is expressed in cultured mesenchymal stromal cells, fibroblasts, and pericytes. Additionally, Meflin is found on stromal cells distributed throughout the bone marrow and on pericytes and perivascular cells in multiple organs [54].

The complement-secreting CAFs (csCAFs), located near malignant cells, particularly in early PDAC, express system complements and were recently confirmed to be distinct from iCAFs, albeit with overlapping signatures [55,56].

Finally, other CAF subpopulations where identified in other tumor types, such as vascular CAFs (vCAFs), found in breast cancer, and cholangiocarcinoma, which highly expressed microvasculature-associated genes (e.g., CD146), as well as inflammatory chemokines such as CCL8 and IL-6 [57,58]. scRNAeq also identified the presence of CD63+ CAFs in breast and prostate cancer, as well as Ptgs2-expressing fibroblasts (RPFs), found in mouse intestinal mesenchyme and in healthy human colons, near the stem cell zone at the bottom of the crypts, where intestinal tumors are primarily initiated [59,60,61]. The presence of CAFs subtypes not found in PDAC reflects the inter-tumoral heterogeneity of CAFs.

### 2.4. CAFs Functional Heterogeneity

Stromal fibrosis and stiffening induced by the ECM produced by CAFs, particularly prevalent in aggressive cancers like PDAC, also enhance cancer cell malignancy and resistance to therapy [62,63]. Nevertheless, studies focusing on CAFs yielded conflicting findings. Genetic depletion of proliferating α-SMA+ CAFs and conditional inhibition of the Sonic Hedgehog (SHH) signaling pathway, vital for desmoplasia promotion in PDAC, resulted in PDAC progression in mouse models. This suggests that specific CAF populations might exert a suppressive effect on cancer progression [64]. Furthermore, clinical studies using CAFs inhibitors were negative, raising the already discussed question whether CAFs promote or restrain cancer progression [30]. Mizutani et al. suggested that CAFs can be divided into cancer-promoting CAFs (pCAFs) and cancer-restraining CAFs (rCAFs) [53].

#### 2.4.1. The Tumor-Promoting Functions of CAFs

One of the cancer’s hallmarks consists of persistent proliferation, where cancer cells can stimulate proliferation through autocrine and interact reciprocally with other cells in the TME to form feedback signals [65]. Among them, the cross-linking signalling between CAFs and tumor cells has been extensively demonstrated [66]. In the TME of PDAC, CAFs express paracrine molecules that promote tumor growth such as growth factors, chemokines, and cytokines [62,64]. For example, SHH signalling exhibits heightened activity in PDAC, inducing insulin growth factor 1 (IF1) and growth arrest-specific (GAS6) expression, both activating Akt signalling in the tumor cell, leading to increased cell proliferation and resistance to apoptosis. Additionally, SHH overexpression also promotes desmoplastic reaction; in a murine PDAC model, tumors deficient in SHH displayed reduced stromal content but increased aggressiveness, heightened vascularity, undifferentiated histology, and cell proliferation [63,64]. Similarly, TGF-β in PDAC leads to the transformation of quiescent fibroblasts to transform into CAFs, which, in turn, secrete TGF-β, promoting tumor cell growth and ECM deposition [67]. 

Another prominent characteristic of cancer is the angiogenesis [65], and scRNA-seq in PDAC confirmed that CAFs overexpress multiple proangiogenic factors, supporting their pro-angiogenic role [68]. By recruiting myeloid cells and monocytes and attracting vascular endothelial cells, CAFs facilitate the tumor-associated blood vessels’ growth and accelerate angiogenesis [30].

Tumor metastasis is another stage in tumorigenesis where CAFs are involved. Fibronectin (Fn), a large extracellular matrix protein, plays a crucial role in cell adhesion, regulating cell polarity, and differentiation. CAFs align Fn through increased contractility and traction, promoting the directed migration of prostate and pancreatic cancer cells, which are mediated by α5β1 integrins and PDGFRα [69]. Furthermore, myCAFs were found to trigger the metastasis of PDAC by inducing type III collagen hyperplasia through the IL-33-ST2-CXCL3-CXCR2 axis. Metastatic CAFs exhibited higher expression of heparan sulfate proteoglycan 2 (HSPG2) or perlecan, known for their pro-metastatic functions, compared to weakly metastatic cancer cells. Notably, CAFs derived from mutant-educated cells (KPflC and KPC mice) created a microenvironment that facilitated invasion [70,71,72,73].

Several studies have confirmed that CAFs play a role in chemotherapy and radiotherapy resistance through various mechanisms, ultimately contributing to treatment failures [4]. In PDAC, the ECM creates a physical barrier that impedes the penetration of chemotherapy and immunotherapy drugs to the cancer site. This barrier compresses peripheral blood vessels, reducing blood flow and, consequently, diminishing the effectiveness of drug delivery [30]. On the flip side, CAFs can bolster tumor cell resistance by directly releasing cytokines and transmitting exosomes. In PDAC, CAF-secreted CXCL12 contributed to tumor progression and gemcitabine resistance via upregulating SATB-1 secretion [74]. CAFs exhibited intrinsic resistance to gemcitabine when compared to normal fibroblasts [74]. Additionally, CAFs transmitted miR-106b directly to pancreatic cancer cells through exosomes, thereby targeting the TP53INP1 gene and fostering gemcitabine resistance in cancer cells [75]. 

In PDAC, CAFs play a dual role in immune modulation. First, they hinder immune cell infiltration by generating a dense fibrotic stroma. This stroma is composed predominantly of immunosuppressive cells such as dendritic cells (DCs), tumor-associated macrophages (TAMs), myeloid-derived suppressor cells (MDSCs), regulatory T cells (T-regs), and cytotoxic T cells [71]. Pancreatic CAFs secrete various factors to recruit and regulate these cells, promoting an immunosuppressive microenvironment. TAMs, MDSCs, and T-regs, in turn, suppress anti-tumoral responses, contributing to tumor growth [76]. Additionally, certain activated fibroblasts express FAP-α, which has the capability to cleave type I collagen (Col 1). This enzymatic activity contributes to an increase in macrophage adhesion, potentially influencing the interactions between fibroblasts and immune cells in the tumor microenvironment [77]. Additionally, CAFs play a part in immune evasion by expressing immune checkpoint ligands, like CTL-associated antigen 4 (CTLA-4) and programmed death-ligand 1 (PD-L1) that bind effector T cells, prompting their dysfunction, but also by releasing suppressive cytokines and chemokines, such as IL-6, IL-1β, CXCL1, CXCL2, and CXCL12 [78,79]. Additionally, CXCL12, secreted by FAP+ CAFs, further inhibits the accumulation of cytotoxic T cells near the tumor, contributing to immune evasion in PDAC [80]. However, it is important to recognize that various subtypes of CAFs may fulfill distinct functions in this mechanism.

In PDAC, the fibrotic stroma imposes restrictions on nutrient and oxygen availability [30,81]. To adapt and survive, pancreatic cancer cells undergo metabolic reprogramming, shifting from oxidative phosphorylation (OXPHOS) to aerobic glycolysis, a phenomenon known as the Warburg Effect. Interestingly, PDAC cells can exploit nearby CAFs to secure energy and nutrients. This phenomenon, termed the “Reverse Warburg Effect”, involves inducing metabolic changes in CAFs similar to aerobic glycolysis [82]. CAFs, under this influence, release energy-rich metabolites like lactate and pyruvate. These metabolites are then absorbed by cancer cells, fueling OXPHOS and facilitating efficient energy production [83]. Furthermore, CAFs enhance glycolytic metabolism by secreting paracrine hepatocyte growth factor (HGF) [84]. By means of autophagy, CAFs can supply alanine as an alternative carbon source, thereby supplying tumor metabolism and growth [85]. Alanine competes with glucose and glutamine, sustaining OXPHOS, non-essential amino acid synthesis, and lipid biosynthesis in PDAC [86]. In addition to directly providing metabolites, CAFs play a role in nourishing tumors by generating ECM rich in nutrients. For example, cancerous cells can absorb extracellular collagen as a source of proline [87]. Similarly, hyaluronic acid within the ECM can act as a nutritional resource for PDAC metabolism [88]. Furthermore, research employing carbon-13-labeled metabolites shows that tumor cells take up CAF-derived exosomes via a macropinocytosis-like process, thereby supplying carbon sources like amino acids and lipids [89].

#### 2.4.2. The Tumor-Restraining Functions of CAFS

Aligned with the previously discussed functional heterogeneity of CAFs, it is important to note that despite the predominant focus of existing studies on the diverse pro-tumor functions of CAFs, their inhibitory roles in cancer should not be overlooked. 

The removal of α-SMA+ myCAFs in PDAC hindered tumor immune surveillance, resulting in an elevated percentage of regulatory T cells (Treg, CD4+Foxp3+). This shift contributed to aggressive tumor progression and a decrease in overall survival [27,63,68]. Furthermore, Rhim et al. identified SHH as a soluble ligand overexpressed in PDAC tumor cells, promoting the formation of a fibroblast-rich stroma. Intriguingly, deleting SHH in murine models reduced tumor interstitium, but these SHH-deleted tumors exhibited heightened proliferative ability and aggressiveness [63]. This suggests that myCAFs play a tumor-suppressive role. Simultaneously, Bhattacharjee et al. demonstrated that myCAF-expressed type I collagen acts as a physical constraint on desmoplastic tumor growth, suppressing tumor growth by mechanically constraining tumor spread, partially mediated through the SHH-SMO signaling pathway. Their findings revealed that the deletion of type I collagen in murine models significantly enhanced tumor metastasis in both PDAC and colorectal cancer metastasizing to the liver [90]. In line with the aforementioned findings, Chen et al. performed the deletion of type I collagen in α-SMA+ myofibroblasts in a PDAC mouse model, resulting in a significant reduction in the overall survival of mice and an acceleration of PDAC progression [91]. Moreover, apCAFs have been reported to present antigens to CD4+ T cells, implicating their involvement in the anti-tumor process. Further evidence is awaited to fully elucidate their anti-tumor role [9]. Similarly, iCAFs have been related to favorable outcomes [92,93,94]. In a recent study, high circulating levels of Osteoglycin (OGN) were identified as a candidate serum prognostic marker derived from a subgroup of iCAFs, associated with a favorable overall survival in resectable PDAC [93].

In summary, in the intricate landscape of PDAC, the dual role of CAFs, simultaneously contributing to both pro-tumoral and anti-tumoral effects, takes center stage. This nuanced interplay underscores the profound influence CAFs wield in shaping the dynamic trajectory of PDAC progression, marking a critical aspect within this complex microenvironment.

### 2.5. CAFs Heterogenous Spatial Distribution 

Another interesting aspect of CAF heterogeneity is their divergent spatial localization in the PDAC microenvironment.

CAFs were found to be spatially divergent, as myCAFs have been found to be proximal to the cancer cells, whereas iCAFs are distant from the cancer cells [12]. In fact, myCAFs were observed in the periglandular region in vivo, again verifying that the formation of myCAFs requires direct interaction with cancer cells in PDAC [12]. 

In a recent study, this heterogeneity was characterized in resected PDAC samples and biopsies from metastatic PDAC lesions through the large-scale integration of histology-guided regional multi-OMICs [11]. Grünwald et al. categorized the PDAC TME in what they termed the subTME, delineating histological features as follows: the “deserted” TME, characterized by an acellular appearance and thin, spindle-shaped fibroblasts; and the “reactive” TME, featuring active fibroblasts and abundant inflammatory infiltrate. An intermediate TME state between these two was also discernible. Additionally, significantly different gene and protein expression signatures among these TME variants were identified through combined proteomics and transcriptomics analysis. Intriguingly, CAFs from these subTMEs exhibited distinct transcriptional signatures and functional phenotypes. In vitro, CAFS from deserted TMEs conferred chemoresistance to cancer cells, while CAFs from the reactive TME supported the proliferation of the basal/squamous-type cancer cells [11]. Although the transcriptional signature of these CAF populations did not fully align with the myCAF–iCAF axis, distinct transcriptional programs were still evident, with CAFs from deserted TMEs enriched in cell cycle genes and CAFs from the reactive TME showing upregulation of inflammation-associated genes [11]. In another comprehensive study, Croft et al. employed spatial transcriptomics and scRNA-Seq datasets to delineate the transcriptomic landscape of CAFs in both tumor-proximal and tumor-distal regions. Tumor-proximal fibroblasts, primarily myofibroblasts expressing podoplanin (PDPN) and enriched in Wnt ligand signaling, were identified. In contrast, tumor-distal subsets were dominated by inflammatory CAFs expressing complement components and the Wnt-inhibitor-secreted frizzled related protein 2 (SFRP2) [92]. Clinical outcomes were correlated with distinct fibroblast profiles; poor outcomes were associated with elevated hypoxia-inducible factor (HIF)-1α and PDPN, while extended survival was linked to inflammatory and complement gene expression [92]. The study underscores the remarkable transcriptional heterogeneity of CAFs, influenced by their spatial proximity to the tumor, providing crucial insights into the complex interplay between CAFs and clinical prognosis. 

### 2.6. CAFs Temporal Heterogeneity 

CAFs heterogeneity can also exhibit temporal dynamics. For instance, myCAFs have been demonstrated to adopt different phenotypes at various stages of PDAC mouse models. In late PDAC progression, one CAF subpopulation diminished in comparison to two other major CAF subsets expressing genes associated with growth factor signaling, inflammation, and myofibroblast markers Acta2 and Tagln [49]. Similarly, myCAFs are implicated from the early stages of tumorigenesis: scRNA-seq revealed the presence of myCAFs in both samples derived from PDAC samples and human intraductal papillary mucinous neoplasia (IPMN)—a common cystic precursor lesion of PDAC [95]. In addition, the authors found that myCAFs were highly represented in high-grade IPMNs but rare in low-grade IPMNs, which also means that myCAFs are present in non-invasive neoplasia [95]. The same study showed that iCAFs were abundant in PDAC but absent in IPMN, which might show that iCAFs promote IPMN transformation into PDAC.

### 2.7. CAFs Heterogeneity and Prognosis

CAF differentiation is determined by the absence of markers associated with epithelial, hematopoietic, and endothelial cells. Additionally, the assessment of protein levels using techniques such as flow cytometry and IHC has revealed specific markers such as α-SMA, FAP, platelet-derived growth factor receptor (PDGFRβ), and fibroblast-specific protein 1 (FSP1), NG2, Integrinβ1/CD29, and Caveolin1/CAV [96]. In contrast to normal fibroblasts, where these proteins are either undetected or present at very low levels, they accumulate distinctly in CAFs [97]. 

Increased stromal FAP expression, a serine peptidase, is linked to aggressive behavior in epithelial cancers and poor prognosis. In breast cancer, FAP+ CAFs enhance invasiveness and diminish anti-tumor immune response, contributing to reduced patient survival [97]. Four CAF populations (named CAF-S1 to S4) have been identified based on the expression of the above-mentioned markers in breast and ovarian cancer [97]. The predominant CAF-S1 (FAPHigh, SMAMed–High, FSP1Med, PDGFRβMed–High, CD29Med, and CAV1Low) and CAF-S4 (FAPNeg–Low, SMAHigh, FSP1Low–Med, PDGFRβLow–Med, CD29High, and CAV1Low) have been observed in other tumor types including PDAC and colorectal cancer [97]. Gene expression analysis for CAF-S1 unveiled enrichment in ECM and inflammation, while CAF-S4 exhibited a perivascular signature [97]. Given the pro-metastatic and immunosuppressive functions of CAF-S1, FAP emerges as an attractive strategy for visualizing and therapeutically addressing CAF-S1 [97]. As previously mentioned in PDAC, two FAP+ CAF populations have been identified: FAP+ SMAhigh (myCAFs) and FAP+ SMAlow (iCAFs). Clusters within FAP+ CAF subpopulation were identified across several tumor types, and the inter-tumor and inter-patient heterogeneity was proven [97].

Although high expression of α-SMA (myCAFs) is predictive of poor survival in PDAC patients, a depletion of α-SMA-expressing cells resulted in immunosuppression and decreased survival in mice [68,98]. Yet, in another study, changes in circulating LIF levels, which induce an inflammatory CAF state, were closely related to tumor response to treatment [46,50]. IL33 is specifically elevated in PDAC and stimulates the CXCL3–CLXCR2 axis to convert CAFs into myCAFs, which are also inversely associated with survival in patients with PDAC [70]. LIF and IL33/CXCL3 could be used as circulatory prognostic markers in PDAC. 

Decorin (DCN) and PDPN have been identified as pan-CAF markers across all three primary CAF populations, i.e., myCAFs, iCAFs, and apCAFs, representing potential candidates for exclusive CAF markers [29]. A meta-analysis comprising 4000 patients with solid tumors from 29 studies found that PDPN+ CAFs were associated with poor survival rates, indicating that PDNP+ CAFs are a valuable prognostic marker [99]. Similarly, meCAFs are linked to unfavorable prognosis. Intriguingly, the presence of meCAF in PDAC patients has a positive impact on the response to protein death 1 (PD-1)-targeted immunotherapy. The enhanced response to immunotherapy in patients with abundant meCAFs may stem from the loose ECM, which facilitates immune cell infiltration. In addition, the authors propose a direct interaction between meCAFs and T cells [52]. Similarly, Dominguez et al. found that myCAFs highly expressing a leucine-rich repeat containing 15 (LRRC15) protein surrounded tumor islets in PDAC and were not present in normal pancreatic tissue [10]. A clinical trial involving more than 600 patients with different tumor types showed that elevated levels of LRRC15+ myCAFs signaling were associated with poor outcomes after anti-PD1 therapy. As such, LRRC15+ myCAFs could be a prognostic marker for immune blockade therapy in PDAC [10]. Conversely, Meflin+ CAFs correlate with a favorable outcome in PDAC patients and mouse models. Meflin+ CAFs have the capacity to suppress α-SMA expression (myofibroblastic differentiation) in CAFs, therefore inhibiting structural remodeling and crosslinking of collagens, which is essential to crucial progression. As such, Meflin+ CAFs favor a less aggressive TME, and could be identified as rCAF subtype, underlining the importance of defining CAF subpopulations as not all CAFs should be regarded as tumor promoting [53].

In an effort to pinpoint prognostic biomarkers, Erkan et al. illustrated that in patients with upfront resected PDAC, those with high collagen but low PSC activity exhibited a favorable prognosis. They introduced a marker termed the activated stroma index (ASI), which represents a ratio of activated CAFs to collagen deposition [98]. Consistent with these findings, Heger et al. validated that a low ASI, indicating low myCAF density and/or high stromal collagen deposition, conferred a favorable prognostic factor in 69 resected PDAC patients. Interestingly, after neoadjuvant treatment with FOLFIRINOX, the significance of ASI seemed to be reversed, and high ASI values were associated with improved survival [100]. Furthermore, the same study showed distinct properties of the myCAF and collagen compartment in PDAC tumor stroma after neoadjuvant treatment with FOLFIRINOX and gemcitabine with nab-paclitaxel, and their differential association with survival compared with stroma without neoadjuvant treatment was assessed. These divergences preclude the use of ASI after neoadjuvant treatment and underscore the diverse and heterogeneous nature of the stroma, influenced by factors such as staging and the type of treatment administered [100]. These findings require confirmation through prospective studies.

### 2.8. CAFs Plasticity

Plasticity is a key characteristic of CAFs and could contribute to their heterogeneity as CAF populations can derive from each other in the presence of a specific stimuli. Biffi et al. showed that IL1/JAK–STAT and TGFβ/SMAD2/3 are two opposing signaling pathways that induce iCAF or myCAF formation, respectively [46]. It is also evident that myCAFs and iCAFs are interchangeable and interconvertible in vitro, depending on their location and exposure in the tumor [12,46]. Furthermore, a small proportion of α-SMA/pSTAT3 double-positive CAFs were also identified, which might represent a transitional state between the iCAFs and myCAFs, further sustaining the theory of CAFs plasticity [46]. Similarly, in human tumors, apCAFs were able to convert into myCAFs, sharing the plasticity shown for other CAF subpopulations [9].

## 3. FAPI PET Imaging in Pancreatic Cancer

### 3.1. Introduction to FAP-Targeted Radiopharmaceuticals

A type II transmembrane serine protease, FAP, is highly expressed on CAFs, particularly in tumors exhibiting substantial desmoplasia like pancreatic cancer. The consistent overexpression of FAP on CAFs distinguishes them from normal fibroblasts [101]. This selective and extensive expression of FAP on CAFs makes it an appealing target for both imaging and therapeutic interventions across various tumors.

Clinical trials conducted with the I-131-labeled monoclonal murine antibody mAb F19 [102] and its humanized version, sibrotuzumab [103], validated FAP as a target for molecular imaging, following the demonstration of highly selective tumoral expression pattern that allows for small lesion detection in patients with colorectal cancer [103]. However, the applications of these radiolabeled antibodies and peptides targeting FAP in nuclear medicine faced certain limitations due to the prolonged circulation and slow clearance caused by their high molecular mass, prompting the introduction of small molecules. 

In 2014, the University of Antwerp developed UAMC-1110, which is a highly potent small-molecule FAP inhibitor with low nanomolar FAP affinity and high selectivity over related enzymes [104,105]. The Haberkorn group at the University of Heidelberg subsequently designed FAPI (FAP inhibitor) precursors and various FAPI tracers based on this motif [106,107]. They developed many FAPI variants in order to improve the potential therapeutic efficacy by increasing tumor uptake, resulting in higher dose delivery. From the 15 different FAPI tracers synthesized by Linder and colleagues, FAPI04 was initially identified as the most suitable potential theranostic tracer [107]. However, since this agent’s pharmacokinetics were still suboptimal for radionuclide therapy, Loktev and colleagues developed 11 further FAPI derivatives. Of those compounds, FAPI46 proved to be more favorable than others since it displayed considerably longer retention than others [108]. In addition to these, several research groups have introduced a variety of FAP-targeting ligands, including an ultra-high-affinity ligand known as oncoFAP [109], a cyclic peptide called FAP-2286 [110], and homodimeric molecules DOTA(SA.FAPi)2 and DOTAGA.(SA.FAPi)2 [111,112]. FAPI04 and FAPI46 are the most commonly used FAPI agents in the literature currently, and they are labeled with the radionuclide Gallium-68 (Ga68). However, Ga68-labeled tracers have several disadvantages due to the limited batch production of Ge68/Ga68 generators and the relatively short-life of Ga68 (68 min), resulting in limited delivery to remote centers. Furthermore, taking into account the lower positron energy of Fluorine-18 (F18), leading to a shorter positron range and higher spatial resolution than Ga68, FAPI molecules radiolabeled with F18 have also been developed [113,114]. The most commonly used F18-labeled tracers are FAPI74 and FAPI42.

Since the initial introduction of FAP-targeted compounds, the primary challenge in the successful implementation of radionuclide therapy has been the short tumor retention time of the molecules. Over time, efforts have been made to extend the tumor retention time of the ligands. Additionally, alternative approaches, such as dimerization (e.g., DO-TA(SA.FAPi)2 and DOTAGA.(SA.FAPi)2) [111,112], albumin binding (e.g., Evan’s Blue conjugates) [115,116], and diverse classes of molecules (e.g., cyclic peptide FAP-2286) [110,117], have been explored to address this issue. 

There are many FAP-targeted radiotracers that have been developed and are under development to be used in imaging and therapeutic applications. Preclinical and clinical studies with these compounds are still ongoing.

### 3.2. Advantages of FAPI PET

Currently, PET/CT stands as a widely used imaging modality in clinical oncology. A glucose analog, fluorodeoxyglucose (FDG), labeled with F18, is the dominant tracer in identifying malignancies on PET/CT scans. Nonetheless, FDG PET/CT has certain limitations in particular indications, including the diagnosis and staging of pancreatic cancer. First, it can occasionally yield false-negative outcomes, especially for the detection of small pancreatic cancers. Second, FDG PET/CT demonstrates relatively low-to-moderate sensitivity in assessing metastatic lymph nodes [118,119], leading to an underestimated N stage and restricting its effectiveness for surgical planning in pancreatic cancer patients. Moreover, its performance in detecting liver metastases and peritoneal carcinomatosis—common forms of pancreatic cancer metastasis—is suboptimal [119]. The latter paragraph explains how FAPI PET can overcome these FDG PET shortcomings.

The principal advantage of FAPI PET over FDG PET lies in its minimal or absent up-take in normal organs. This characteristic leads to better detection of liver metastases and peritoneal carcinomatosis due to the lack of tracer uptake in the liver and intestines. The low uptake in healthy tissues and heightened uptake in malignant tissues contribute to a distinct contrast between the tumor and its surroundings, facilitating precise tumor delineation. Additionally, considering that the stromal component comprises a substantial majority of the pancreatic tumor mass [24], stroma-targeting FAPI PET is expected to be more sensitive than tumor-cell-targeting FDG PET, especially in small lesion detection. Consequently, the improved tumor delineation and high sensitivity in targeting small lesions could potentially enhance the detection of small pancreatic tumors and lymph node metastases.

Furthermore, FAPI PET presents advantages over FDG PET regarding patient preparation. Unlike FDG PET, it does not necessitate fasting or avoidance of strenuous exercise 24 h prior to scanning, nor does it necessitate a pause of insulin administration or the need for a warm and quiet environment post-injection [120]. Moreover, early imaging—for instance, within 10 min post-injection—is feasible [120,121,122], which significantly reduces the patient waiting time. These practical advantages make FAPI PET much more feasible, particularly for diabetic and critical care patients.

### 3.3. Clinical Applications of FAPI PET in Pancreatic Cancer

#### 3.3.1. FAP Expression and Its Correlation with FAPI PET Uptake in Pancreatic Cancer

In pancreatic cancer, FAP is expressed not only on CAFs but also on the tumoral cells themselves [123]. Given the high proportion of desmoplastic stroma in pancreatic cancer and tumor cell FAP expression, it is expected to show intense FAP expression. Consistently, Mona et al. reported strong FAP expression in 50–100% of pancreatic cancer cases using FAP IHC scoring [124]. Correlated with the high FAP expression in tissue, high uptake on FAPI PET has been demonstrated in pancreatic cancer [20,125]. Unsurprisingly, Kessler et al. reported a significant moderate correlation of maximal standardized uptake values (SUVmax) on Ga68-FAPI PET and histopathologic FAP expression (immunohistochemical FAP score) in 18 pancreatic cancer samples (r = 0.60, *p* < 0.01) [126], similarly as in Ding et al. (r = 0.78, *p* < 0.05) [125]. Furthermore, Spektor et al. reported a concordance between FAP expression and SUVs on FAPI PET in 14 low-grade and high-grade IPMN cases, both increasing with the malignant transformation [127].

#### 3.3.2. Comparison of FAPI PET and FDG PET Diagnostic Performance in Pancreatic Cancer

The comparison of FAPI PET with FDG PET is intriguing since it is a promising PET tracer for pancreatic cancer imaging. Therefore, most of the studies compared these two PET scans in terms of the uptake and detectability of primary and metastatic pancreatic cancer lesions. Regarding primary pancreatic lesions’ uptake, all the studies agreed on the superiority of FAPI PET over FDG PET [20,126,127,128,129,130,131,132,133,134] (Figure 2). SUVs of primary lesions, the most commonly utilized quantitative parameter of PET scans, were found to be significantly higher on FAPI PET than FDG PET [20,126,127,128,129,130,131,132,133,134]. Consistently, FAPI PET sensitivity of primary tumor detection also surpassed the sensitivity of FDG PET [20,126,129,130,131,132,134]. On the other hand, Pang et al. and Kessler et al. noted that FAPI PET showed lower specificity in identifying primary lesions compared to FDG PET [20,125]. The false-positive cases were higher on FAPI PET scans. The reduced specificity of FAPI PET in primary tumor detection can be attributed mainly to an overlap in uptake intensity in the pancreatic primary tumor and in the tumor-induced obstructive pancreatitis of the pancreatic parenchyma [134]. Dual-time-point (3 h delayed) imaging has been investigated as a solution to distinguish between tumors and inflammation-induced fibrosis, and positive outcomes have been observed [20,21,135]. 

Comparing the tracer uptake of pancreatic cancer lymph node metastasis on FAPI PET and FDG PET, similarly to the primary tumor lesions, SUVs were found to be significantly higher on FAPI PET [20,126,129,132,133,134]. Predictably, the detection rate of FAPI PET in lymph node metastasis surpassed FDG PET [129,131,132,134], although the difference was not significant in some studies [20]. 

Regarding the distant metastasis in PDAC, Ding et al. and Xu et al. reported significantly higher SUVs on FAPI PET than on FDG PET [132,134]. However, some other studies reported a non-significant superiority in favor of FAPI PET uptake values [126,133] or the absence of a difference between the two PET scan uptakes in total metastatic lesions [130]. Ding et al. noted that the sensitivity and accuracy of metastatic lesion detection were higher on FAPI PET than that of FDG PET [132], which was recently confirmed by Li et al. and Xu et al., particularly in liver and peritoneal lesion detection [133,134]. Consistently, Kessler et al. reported similar findings and emphasized the higher detection rate on FAPI PET compared to both FDG PET and CeCT (contrast-enhanced CT), especially in liver and peritoneal lesions [126].

#### 3.3.3. Prognostic and Predictive Value of FAPI PET in Pancreatic Cancer

FAP expression has been shown to correlate with the clinical outcome. Ogawa et al., using whole-tissue slides from 215 treatment-naïve PDACs, reported that FAP-dominant fibroblast-rich stroma was associated with decreased survival compared to collagen-rich stroma [136]. Similarly, Shi et al. demonstrated that FAP expression was correlated with shorter patient survival and served as an independent prognostic indicator for PDAC [123]. Therefore, the parameters derived from FAPI PET were thought to be associated with the clinical outcome in pancreatic cancer. In a retrospective study involving 37 patients, Ding et al. highlighted the prognostic value of FAPI PET in resectable PDAC. Their findings revealed that SUVmax and TPF (total pancreatic FAP expression) on FAPI PET were an independent negative prognostic factor for, respectively, recurrence-free survival and overall survival (OS) [125]. In a recent prospective study, Zhu et al. analyzed baseline FAPI PET variables in 37 inoperable PDAC patients, identifying metabolic tumor volume (MTV) as an independent predictor of OS [137]. In the same study, they also examined the potential of FAPI PET variables to predict therapy response and survival by assessing changes before and after one cycle of chemotherapy in 17 inoperable PDAC patients. They observed greater alterations in SUVmax, MTV, and TLF (total lesion FAP expression) in patients showing good response compared to those with poor responses, which may have clinical relevance in identifying the risk of disease progression [137]. On the other hand, Li et al. investigated the correlation between SUVmax and TBR on FAPI PET and the response in a cohort of 48 patients receiving systemic treatment, but their findings did not reveal any significant association between these parameters in the response and non-response groups [133].

#### 3.3.4. Impact of FAPI PET in Pancreatic Cancer Staging and Management

Comprehensive evaluations of a patient’s disease condition alongside tailored treatment plans could enhance survival rates. This is particularly crucial for pancreatic cancer patients as less than 20% are immediately eligible for surgery, highlighting the critical role of neoadjuvant therapies in their care. Since FAPI PET has the potential for better detection of primary and metastatic lesions compared to FDG PET and conventional imaging, it may provide useful additional information in the oncologic management of pancreatic cancer patients. 

In a study of 19 PDAC patients, Röhrich et al. investigated the impact of FAPI PET on therapeutic management compared to standard-of-care imaging by CeCT. They reported that FAPI PET led to a change in clinical TNM staging in 10 out of 19 patients (53%) and in oncological management in 7/19 patients (37%) [21]. In a subsequent report covering a larger cohort of 77 pancreatic cancer patients, the same group, Koerber et al., highlighted major TNM upstaging in 26 patients and downstaging in 6 patients, resulting in a total of 7 major and 23 minor modifications in patient management compared to gold-standard imaging (CeCT or MRI) [138]. In another study, Pang et al. found that FAPI PET was superior to FDG PET in terms of TNM staging, causing TNM upstaging in 6 out of 23 patients (26%). The impact on treatment change was less significant: in 2/23 patients (9%) compared to FDG PET and in 1/23 patients (4%) compared to CeCT [20]. Similarly, in their comprehensive cohort of 49 patients, Ding et al. showed that FAPI was superior to FDG in TNM staging (accurately evaluated in 75.5% (37/49) of the patients with FAPI PET and 55.1% (20/36) of the patients with FDG PET) [132]. Comparison of FAPI PET to CeCT demonstrated more lesions on FAPI PET for N and M staging; however, it had erroneous size evaluations on T staging. Eventually, they noted treatment change in nine patients compared to CeCT (18.4%) and in four patients compared to FDG PET (8.1%) [132]. Likewise, Li et al. reported 14 upstaging with FAPI PET among 62 pancreatic cancer patients compared to FDG PET, accompanied by significant improvements in N and M staging compared to both FDG PET and CT/MRI. However, in terms of T staging, while FAPI PET and FDG PET outcomes were similar, vascular involvement assessment was more accurate on CeCT than that of PET/CT, and T4 staging proportion was significantly higher on CT/MRI [133]. Comparing FAPI PET with FDG PET, Lyu et al. reported N-stage improvement in 16/31 patients (51.6%) with FAPI PET, followed by treatment change in 12 patients (38.7%) from surgically resectable to unresectable [131]. Similarly, in a study evaluating the impact of FAPI PET compared to FDG PET, upstaging was evident in 4 out of 22 patients (18%), particularly in progressive and recurrent disease settings, accompanied by 17% alteration in patient management after the introduction of FAPI PET [134]. The least impact of FAPI PET on therapeutic management was noted by Kessler et al.; in their cohort of 59 patients, major changes were documented in 3 patients and minor changes in 2 patients (total 5 patients, 8.5%) when the authors compared the decisions made before and after FAPI PET results [126]. In conclusion, FAPI PET has been shown to be beneficial to pancreatic cancer patient management compared to CeCT/MRI and FDG PET, especially in N and M staging, resulting in therapeutic modifications.

Furthermore, another potential impact of FAPI PET in the therapeutic management of pancreatic cancer may be the improvement of gross tumor volume (GTV) delineation for radiotherapy planning. According to Liermann et al., FAPI PET enables GTV contouring in locally recurrent pancreatic cancer patients, yielding favorable outcomes in seven cases compared to manually contoured target volumes determined on CeCT, which serves as the reference standard for target volume delineation. They concluded that FAPI PET can serve as an additional imaging modality to enhance decision-making in target definition, especially in inconclusive cases [139]. Additionally, Koerber et al., in their cohort of 77 pancreatic cancer patients, where 7 major and 23 minor changes in patient management were observed, emphasized that the greatest impact in patient management was onradiation therapy planning. The authors declared that in their cohort, FAPI PET imaging contributed to enhanced target volume delineation, leading to reduced exposure of organs at risk and improved definition of target volume [138].

#### 3.3.5. FAPI PET in the Discrimination of Suspicious Pancreatic Lesions

IPMN lesions are pathologically categorized as having either low-grade dysplasia, associated with a benign prognosis, or high-grade dysplasia, representing a carcinoma in situ that can transform into PDAC. Despite the high sensitivity of MRI and endoscopic ultrasonography, their lack of high specificity makes them challenging for differentiation between low-grade and high-grade IPMNs. FAPI PET has also been investigated for the discrimination of suspicious pancreatic lesions. In a cohort of 25 patients, Lang et al. demonstrated significantly elevated FAPI uptake in high-grade IPMN compared with low-grade IPMN and other benign cystic lesions. Hence, FAPI PET was interpreted to have the potential to avoid unnecessary surgery for non-malignant pancreatic IPMN [140]. Consistently, Rasinksi et al., in another study with 30 histopathologically confirmed pancreatic lesion cases, observed significantly elevated FAPI uptake in malignant lesions compared with benign lesions. It was noted that FAPI PET can accurately discriminate malignant from benign lesions deemed equivocal by conventional imaging [141]. Recently, Spektor et al. revealed a correlation between the mean immunoreactive score of FAP (determined by the intensity and percentage of FAP-positive cells) and the mean SUVmax, SUVmean, and time-to-peak of FAPI PET in pancreatic lesions. All these parameters showed an increase with higher malignant transformation, being lower in low-grade IPMNs, higher in high-grade IPMNs, and the highest in PDAC. Thus, they concluded that increasing expression of FAP in lesions with a higher degree of malignancy matches the expectation of a stronger FAP expression in PDAC and high-grade IPMNs than in low-grade IPMN [127]. This finding supports their earlier observations of elevated SUVs and prolonged time-to-peak in PDAC and high-grade IPMNs compared to low-grade IPMNs, as previously reported by Lang et al. [140].

Table 1 provides a summary of the key findings from the aforementioned studies on FAPI PET in pancreatic cancer.

## 4. Challenges and Future Directions

### 4.1. FAP Expression and FAP-Targeted PET Imaging

FAP expression typically occurs in fetal mesenchymal tissue, but it is selectively upregulated in reactive stromal fibroblasts within epithelial cancers, dermal scars of healing wounds [142], and liver cirrhosis [143]. The majority of FAP is expressed by activated fibroblasts responding to pathological situations, i.e., predominantly CAFs. Certain CAF subpopulations exhibit FAP expression (e.g., myCAFs, iCAFs, CAF-S1); however, FAP expression is low or negative in certain CAF subpopulations (e.g., CAF-S4). Furthermore, some cancer cells (e.g., sarcoma, certain ovarian, and pancreatic cancers) can exhibit FAP expression as well. FAPI PET imaging utilizes certain radiopharmaceuticals targeting FAP; therefore, it demonstrates FAP expression distribution in the explored fields. Hence, FAPI PET cannot differentiate between distinct CAF subpopulations; rather, it shows the FAP-expressing CAFs (FAP+ CAFs) and other FAP-expressing cells.

### 4.2. Future Potential of FAPI PET in Pancreatic Cancer

The comparative analysis of FAPI PET and FDG PET in pancreatic cancer imaging reveals promising prospects for FAPI PET as a superior imaging modality. The robustness of FAPI PET in detecting primary and metastatic lesions, evident through higher SUVs and superior sensitivity, positions it as a valuable tool in the diagnostic landscape. Importantly, based on the preliminary limited outcomes, FAPI PET parameters exhibit potential prognostic value, aiding in predicting responsiveness to chemotherapy, recurrence-free survival, and overall survival. This prognostic significance underscores the clinical relevance of FAPI PET in shaping treatment strategies. Moreover, FAPI PET’s impact on therapeutic decision-making, evidenced by TNM staging modifications and treatment changes, highlights its potential in optimizing patient management. The integration of FAPI PET into oncological care, especially in neoadjuvant therapy planning, could lead to more accurate TNM staging, therapeutic modifications, and improved gross tumor volume delineation for radiotherapy planning, contributing to enhanced survival outcomes for pancreatic cancer patients. Continued research and prospective studies are essential to further solidify the clinical utility and widespread integration of FAPI PET in routine practice.

Several ongoing clinical trials are exploring the application of FAPI PET in pancreatic cancer diagnosis and management to address unanswered questions (Table 2). In a multicentric, prospective trial named FAPI-PANC, initiated at our center (H.U.B.), our primary aim is to investigate the correlation between FAPI PET and the histopathological/molecular characteristics of the TME, focusing on CAFs. The study will also assess the utility of FAPI PET in staging borderline resectable PDAC patients and the potential predictive value of FAPI PET regarding treatment response to systemic and/or locoregional interventions. In another trial, with registration number NCT05262855, FAPI PET will be employed for initial staging, with a follow-up Ga68-FAPI-46 PET scan conducted before the planned surgical resection following neoadjuvant chemotherapy to perform histopathology and immunohistochemistry analyses, allowing for a comprehensive comparison with resected PDAC tumor specimens. Another trial, NCT05275985, aims to assess the value, performance, and impact of FAPI PET/CT on the clinical management of individuals suspected to have pancreatic cancer. NCT05957250 focuses on determining the optimal timing and scan protocol for Ga68-FAPI-46 PET/CT scans, evaluating its accuracy in detecting pancreatic cancer and assessing its ability to monitor the effects of chemotherapy on pancreatic cancer lesions. Lastly, in trial NCT05518903, the researchers aim to establish the sensitivity and specificity of Ga68-FAPI-46 PET for detecting and quantifying CAFs in PDAC. Additionally, they seek to construct, test, and validate a model predicting surgical benefit or futility in potentially resectable PDAC using Ga68-FAPI-46 PET biomarkers in combination with other disease-related biomarkers. These trials collectively contribute valuable insights into the evolving role of FAPI PET in the comprehensive management of pancreatic cancer.

### 4.3. Future Directions: FAP-Targeted Radioligand Therapy

Theranostics in nuclear medicine represents an integrated approach that combines diagnostic imaging and therapeutic interventions, using radiopharmaceuticals tailored to specific molecular targets. The possibility to specifically target FAP-expressing cells using radioligands is an attractive therapeutic option. Recently, FAP-targeted radioligand therapy (FAP-RLT) has become one of the most appealing topics in nuclear oncology.

The current evidence for FAP-targeted RLT is derived only from case series and proof-of-concept studies in various tumor entities. To date, patient data from treatments involving diverse FAP-targeted radiopharmaceuticals, including Lu177-FAPI-04, Y90-FAPI-46, Lu177-FAP-2286, Lu177-DOTA.SA.FAPI, Lu177-DOTAGA.(SA.FAPI)2, and Lu177-EB-FAPI (LNC1004), have been reported [113,115,117,144,145,146], primarily focused on feasibility, biodistribution, dosimetry, and safety with initial signs of efficacy. Notably, these studies have collectively demonstrated the feasibility and safety of FAP-targeted RLT. Clinical efficacy was observed with mostly disease stabilization in a variety of different tumor types, predominantly breast cancer, pancreatic cancer, thyroid cancer, and sarcoma. While prospective study data are pending, encouraging early findings support the need for further exploration. Three phase 1 trials (NCT04849247, NCT05432193, and NCT05723640) and a phase 1/2 trial (NCT04939610) are ongoing (Table 2).

The effectiveness of RLT relies on various factors, including optimizing radiotherapeutic strategies, overcoming intrinsic resistance in tumor cells, and the modulations on the TME. FAP-radioligands, which act through different mechanisms based on FAP expression on CAFs and also on certain tumor cells, deliver ionizing radiation directly to CAF or tumor cells and may affect nearby cells through crossfire effects. The relative contribution of these mechanisms to FAP-RLT efficacy remains unexplored, with responsiveness potentially varying based on factors such as the type of cells expressing FAP, tumor architecture, FAP-radioligand characteristics, and tumor biology. Understanding the molecular and cellular impact of FAP-radioligands on CAF, tumor cells, and immune cells is crucial, particularly when targeting FAP+ CAF alone. CAF’s general radioresistance, the impact of radioligands on the balance of tumor-supportive and tumor-suppressive CAF subpopulations, and their varying effects over time pose a challenge and require further investigation. It should be noted that the anatomical location and spatial organization within a tumor may be important; in cases where CAFs are intertwined with tumor cells or situated around clusters of tumor cells, such as in pancreatic cancer, there may be an opportunity to leverage crossfire effects. However, this strategy might not be viable in tumors where FAP+ CAFs are situated at a greater distance from tumor cells [147].

In a study comparing beta-emitter Lu177-FAPI-46 and alpha-emitter Ac225-FAPI-46 in pancreatic cancer models, slower but longer treatment effects with Lu177-FAPI-46 were observed. The authors attributed this discrepancy to the fact that the target cells, CAFs, were in the stroma, and they exhibited greater tolerance and radioresistance than the tumor cells [148]. In conclusion, there remains a multi-faceted exploration ahead to enhance the treatment effects of FAP-targeted RLT, determining the optimal choice between beta-emitter and alpha-emitter radionuclides, considering shorter half-life radionuclides and exploring combination treatments with therapies directly targeting tumor cells.

FAP-RLT has gained prominence, demonstrating feasibility, safety, and early signs of efficacy across various tumor types, with ongoing phase 1 and 2 trials, while the impact of FAP-radioligands on tumor microenvironment components, particularly fibroblasts, requires further investigation to understand the mechanisms and optimize the therapeutic outcomes, emphasizing the need for a comprehensive understanding at the molecular and cellular levels.

## 5. Conclusions

In the neoadjuvant context of PDAC, unravelling the complexities of fibroblast activation gains particular significance. Studies on CAFs in PDAC employ a range of advanced techniques for comprehensive characterization. Immunostaining, in situ hybridization, flow cytometry, fluorescence-activated cell sorting, and mRNA microarrays initially laid the groundwork. Subsequently, the advent of scRNAseq has played a pivotal role in unveiling the intricate heterogeneity within CAF populations and the necessity for in-depth exploration. Öhlund’s study, utilizing a three-dimensional in vitro co-culture system, and Elyada’s droplet-based scRNAseq approach identified distinct subtypes, including myCAFs, iCAFs, and apCAFs. These techniques enabled a nuanced understanding of their unique molecular signatures and functional contributions. In parallel, recent investigations into CAF heterogeneity utilized snRNAseq to profile PDAC specimens, identifying four distinct CAF programs. These programs exhibited non-specific overlaps with various cross-tissue fibroblast signatures, underlining the necessity for in-depth exploration. The discovery of additional subtypes like metabolic CAFs (meCAFs) and Meflin+ CAFs, along with spatial and temporal heterogeneity, plasticity, and the dual role of CAF, capable of both pro- and anti-tumoral behavior, further underscores the complex nature of CAF populations. This multi-faceted approach enhances our comprehension of CAF biology in PDAC and sets the stage for targeted interventions and therapeutic strategies. While advanced techniques have significantly deepened our understanding of CAF heterogeneity, exploring additional tools is crucial for developing a comprehensive perspective. Particularly noteworthy biomarkers, such as FAP, α-SMA, PDPN, and LRRC15, have provided valuable insights into CAF subtypes. Understanding the role of FAP in the TME is crucial as it exhibits selective upregulation in reactive stromal fibroblasts within epithelial cancers. While certain CAF subpopulations, such as myCAFs, iCAFs, and CAF-S1, express FAP, others like CAF-S4 may have low or negative expression. FAP is also detected in cancer cells, complicating its role as a biomarker. However, to further unravel the intricacies of fibroblast activation, emerging technologies like FAPI-PET offer a non-invasive imaging approach. 

FAPI PET imaging, utilizing FAP-targeting radiopharmaceuticals, provides a broad view of FAP-expressing cells, emphasizing FAP+ CAFs. However, it lacks the ability to distinguish between diverse CAF subtypes. FAPI PET emerges as a promising imaging modality for pancreatic cancer, addressing limitations associated with FDG PET. The correlation between FAP expression and FAPI PET uptake establishes its robust diagnostic potential. FAPI PET’s impact on therapeutic decision-making, including TNM staging modifications, demonstrates its clinical relevance. Its prognostic and predictive value, coupled with its potential role in refining radiotherapy contouring, position FAPI PET as a valuable tool in the comprehensive management of pancreatic cancer. Integrating such cutting-edge tools with existing methodologies will likely enhance our ability to dissect the complexity of CAF biology and pave the way for more effective therapeutic interventions. Further well-designed, comprehensive, prospective studies, preferably including head-to-head comparisons with the reference standard imaging modalities, are required to solidify the clinical utility of FAPI PET in pancreatic cancer management and its place in routine clinical practice.

The future direction of FAP-targeted radioligand therapy (FAP-RLT) combines diagnostic imaging with therapeutic potential, showing promise in various cancers, including breast, pancreatic, and thyroid cancers. Early studies highlight feasibility, safety, and signs of efficacy, with ongoing trials (NCT04849247, NCT05432193, NCT05723640, and NCT04939610) aiming to validate its potential. However, many challenges persist, including developing a better understanding of the exact FAP+ target populations among CAFs, optimizing radiotherapeutic strategies, and addressing CAF’s radioresistance. A comprehensive understanding at the molecular and cellular levels is imperative for leveraging FAP-RLT effectively, emphasizing the need for continued research to optimize therapeutic outcomes in the complex interplay of the TME and fibroblast activation.

## Figures and Tables

**Figure 1 biomedicines-12-00591-f001:**
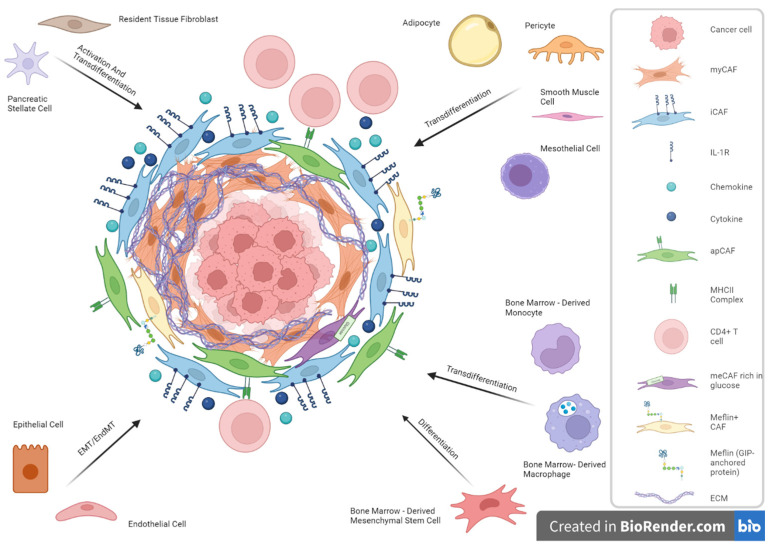
Cancer-associated fibroblasts (CAFs) and cellular origin heterogeneity in Pancreatic Ductal Adenocarcinoma (PDAC).

**Figure 2 biomedicines-12-00591-f002:**
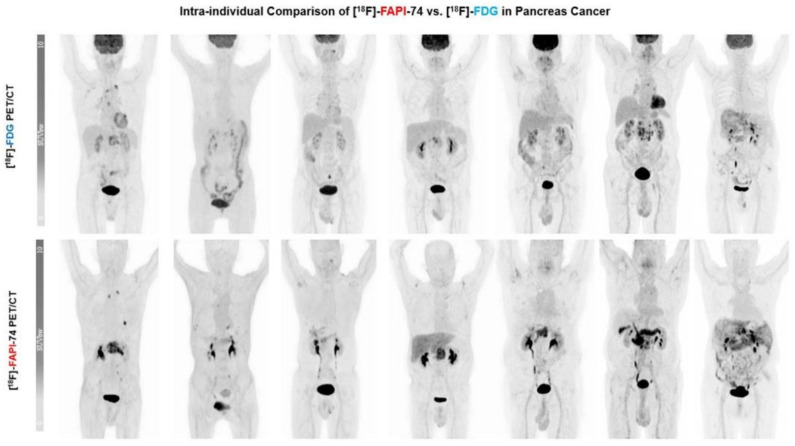
Maximum intensity projections (MIP) in seven patients: F18-FDG vs. F18-FAPI-74. Primary lesions exhibited substantially better image contrast and delineation with F18-FAPI-74. (Figure adapted from Novruzov et al. [130] under a CC BY license; link of the license: https://creativecommons.org/licenses/by/4.0/, accessed on 27 November 2023.).

**Table 1 biomedicines-12-00591-t001:** Summary of the main studies regarding FAPI PET in pancreatic cancer.

Author and Publication Year	Study Design and Total Patient Number	Main Conclusion
Röhrich et al., 2020 [21]	Retrospective, *n* = 19	Re-staging with Ga68-FAPI PET/CT in half of the patients with PDAC and most patients with recurrent disease compared with standard of care imaging.
Liermann et al., 2021 [139]	Retrospective, *n* = 7	FAPI-PET/CT being a superior imaging modality due to its high tumor-to-background contrast compared to the current gold standard contrast-enhanced CT in pancreatic cancer. Demonstration of how FAPI-PET/CT could facilitate target definition and increases consistency in radiation oncology in pancreatic cancer.
Pang et al., 2021 [20]	Retrospective, *n* = 36	Higher sensitivity of Ga68-FAPI PET compared with F18-FDG PET/CT in detecting primary pancreatic tumors, involved lymph nodes, and metastases. Superiority of Ga68-FAPI PET in terms of TNM staging.
Zhang et al., 2022 [128]	Prospective, *n* = 33	Better detectability of suspicious lymph node metastases with Ga68-FAPI-04 PET compared to F18-FDG PET. MR multiple sequence imaging of Ga68-FAPI-04 PET/MR, explaining pancreatic lesions in patients with obstructive inflammation and detecting tiny liver metastases.
Lang et al., 2022 [140]	Retrospective, *n* = 25	Ga68-FAPI PET being a valuable new tool for distinguishing pancreatic IPMN grades, avoiding unnecessary surgery.
Ding et al., 2023 [125]	Retrospective, *n* = 37	Significant correlation between in vivo Ga68-FAPI-04 uptake with ex vivo FAP expression and aggressive pathological characteristics in localized PDAC. Potential postoperative prognostic value of Ga68-FAPI-04 PET/CT in PDAC.
Liu et al., 2023 [129]	Retrospective, *n* = 51	Higher sensitivity and accuracy of Ga68-DOTA-FAPI-04 PET/CT compared to F18-FDG PET/CT in diagnosing pancreatic cancer. Independent prognostic value of Ga68-FAPI PET for pancreatic cancer patients.
Novruzov et al., 2023 [130]	Prospective, *n* = 7	Markedly elevated uptake of F18-FAPI-74 with a lower background uptake, providing a very high visual contrast. Higher number of detected lesions with F18-FAPI-74 compared to F18-FDG PET.
Zhu et al., 2023 [137]	Prospective, *n* = 47	Association of higher baseline MTV on F18-NOTA-FAPI-04 with poorer survival in inoperable PDAC patients. Higher sensitivity of ΔMTV on FAPI PET than ΔCA19-9 for predicting response.
Rasinski et al., 2023 [141]	Prospective, *n* = 30	Accuracy of Ga68-FAPI-46 PET/CT in differentiating malignant from benign pancreatic lesions deemed equivocal by standard-of-care imaging. FAPI PET being a necessary tool when standard-of-care imaging is inconclusive.
Ding et al., 2023 [132]	Prospective, *n* = 49	Higher sensitivity and accuracy of Ga68-FAPI-04 PET/CT than F18-FDG PET/CT for tumor, node, and metastasis staging of PDAC identified on CeCT. Significant association of Ga68-FAPI-04 uptake with pathologically aggressive tumor features. Improvement in prognostic value when Ga68-FAPI-04 and F18-FDG PET/CT findings were combined.
Lyu et al., 2023 [131]	Prospective, *n* = 31	Equivalent detection ability of pancreatic lesion to F18-FDG PET/CT with F18-NOTA-FAPI-04 PET/CT at post-injection 15- and 30-min images. Distinguishing pancreatic carcinoma from tumor-associated inflammation with delayed-phase F18-NOTA-FAPI-04 PET/CT. Better performance of F18-NOTA-FAPI-04 PET/CT in TNM staging compared to FDG PET/CT.
Kessler et al., 2023 [126]	Prospective, *n* = 62	Association of Ga68-FAPI PET SUVmax and histopathologic FAP expression in pancreatic cancer patients. High detection rate and diagnostic accuracy of FAPI PET, superior to those of F18-FDG PET/CT.
Spektor et al., 2024 [127]	Retrospective, *n* = 98	Validation of FAP as a biology-based stromal target for in vivo imaging. Increasing expression of FAP in lesions with a higher degree of malignancy among low-grade IPMN and high-grade IPMN and PDAC.
Li et al., 2024 [133]	Prospective, *n* = 62	Better performance of F18-FAPI-04 PET/CT than F-18-FDG PET/CT in identification of primary tumors, lymph node, and distal metastasis, and in TNM staging of PDAC.

**Table 2 biomedicines-12-00591-t002:** Main ongoing clinical trials studying FAP-targeted imaging and therapy in PDAC.

Study ID	Study Design	Eligibility Criteria	Intervention	Primary Endpoint
Diagnostic clinical trials				
NCT05083247 FAPI-PANC	Prospective sub-study, NR *n*= 30	BR PDAC	Ga68-FAPI PET/CT	Establish a correlation between Ga68-FAPI PET/CT and histopathological and molecular biomarkers
NCT05262855	Phase 2, NR *n* = 60	Resectable or BR PDAC	Ga68-FAPI PET/CT	Sensitivity, specificity, and accuracy to detect FAP-expressing cells using histopathology as true standard
NCT05275985	Prospective, NR *n* = 80	Patients with pancreatic lesions	Ga68-FAPI PET/CT	SUV, number of patients who changed treatment, number of lesions detected, PFS, OS
NCT05957250 PANSCAN-1	Prospective study, NR *n* = 60	Resectable or BR PDAC	Ga68-FAPI PET/CT	Sensitivity, specificity, and accuracy to detect FAP-expressing cells using histopathology as true standard
NCT05518903	Phase 2, NR *n* = 130	Resectable or BR PDAC	Ga68-FAPI PET/CT	Sensitivity, specificity, and accuracy to detect FAP-expressing cells using histopathology as true standard
FAP-targeted Radioligand Therapy Clinical Trials				
NCT04849247	Phase 1, NR *n* = 30	Advanced or metastatic solid tumors	Ga68-DOTA-FAPI 177Lu-DOTA-FAPI	DLT RP2D
NCT05432193 FRONTIER	Phase 1, NR *n* = 30	Advanced or metastatic solid tumors (PDAC, CCA, EC CRC, Melanoma, HNSCC, sarcoma)	Ga68-PNT6555 177Lu-PNT6555	Treatment-emergent AE
NCT05723640	Phase 1, NR	FAP+ advanced or metastatic solid tumors	177Lu-LNC1004	AE DLT
NCT04939610 TablLuMIERE	Phase 1/2, NR *n* = 222	FAP+ PDAC, BC, NSCLC	177Lu FAP 2286 monotherapy and combination with chemotherapy	RP2D
NCT06081322	Phase 1, NR *n* = 29	Advanced PDAC and CCA	177Lu-EB-FAPI	AE Objective response rate

Abbreviations: AE, adverse events; BC, breast cancer, BR, borderline resectable; CCA, cholangiocarcinoma; CRC, colorectal cancer; DLT, dose-limiting toxicity; EC, esophageal cancer; FAP+, fibroblast-associated protein positive; FAPI, fibroblast-associated protein inhibitor; Ga, gallium; HNSCC, head and neck squamous cell carcinoma; Lu, Lutetium; *n*, number of patients, NSLC, non-small lung cancer; NR, non-randomized; OS, overall survival; PDAC, pancreatic ductal adenocarcinoma; PET/TC, positron emission tomography/computed tomography scan; PFS, progression-free survival, R, randomized; RP2D, recommended phase 2 dose; SUV, standard uptake value.

## Data Availability

Not applicable.

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
