# Peer review of "Cancer-Associated Fibroblasts in Pancreatic Ductal Adenocarcinoma or a Metaphor for Heterogeneity: From Single-Cell Analysis to Whole-Body Imaging"

_biomedicines, 2024, doi:10.3390/biomedicines12030591_

Round 1
Reviewer 1 Report
Comments and Suggestions for Authors
In the manuscript, Saúde-Conde et al. offer a comprehensive review of the role of cancer-associated fibroblasts (CAFs) in pancreatic ductal adenocarcinoma (PDAC), focusing on heterogeneity, the potential of fibroblast activation protein (FAP) as a diagnostic and therapeutic target, and the emerging role of 68 Gallium FAP inhibitor (68Ga-FAPI-PET) imaging. Here are some suggestions to enhance the quality and impact of this review:
The review thoroughly covers the complexity of CAFs in PDAC, the promise of FAP as a target, and the novel diagnostic potential of 68Ga-FAPI-PET imaging. It adeptly bridges basic science with clinical application, providing a balanced view of current knowledge and future directions. However, integrating additional elements could further strengthen the manuscript.
1. Ensure that all cited studies are up to date. Given the rapid advancements in this field, including the most recent findings will make the review more valuable and relevant, e.g.
a. Spektor, Anna-Maria, et al. "Immunohistochemical FAP Expression Reflects 68Ga-FAPI PET Imaging Properties of Low-and High-Grade Intraductal Papillary Mucinous Neoplasms and Pancreatic Ductal Adenocarcinoma." Journal of Nuclear Medicine 65.1 (2024): 52-58.
b. Millul, Jacopo, et al. "An ultra-high-affinity small organic ligand of fibroblast activation protein for tumor-targeting applications." Proceedings of the National Academy of Sciences 118.16 (2021): e2101852118.
c. Liu, Yuwei, et al. "Fibroblast activation protein targeted therapy using [177 Lu] FAPI-46 compared with [225 Ac] FAPI-46 in a pancreatic cancer model." European Journal of Nuclear Medicine and Molecular Imaging (2022): 1-10.
2. The paper highlights the diagnostic potential of 68Ga-FAPI-PET. Expanding this section to include a comparison with other diagnostic methods for PDAC, such as MRI or CT, could provide a clearer picture of its relative advantages and limitations.
3. Consider adding more figures and tables to summarize key points, such as a table comparing diagnostic methods.
Comments on the Quality of English LanguageThe English language in the manuscript is generally clear and well-structured.
Reviewer 2 Report
Comments and Suggestions for Authors
Dear Author,
Cancer-associated Fibroblasts in Pancreatic Ductal Adenocarci- 2 noma or a metaphor for heterogeneity: From Single-cell analy- 3 sis to whole-body imaging is interesting topic.
I recommend
1.Rewrite abstract and introduction in consistent flow and clarity.One sentence should connected to next and similarly for paragraph that makes coherent writting.
2.Whats novelty of this topic?
3.I will suggest to add more case study ?
4.Add future prospects?
5.Check for grammer and Engligh?
Comments on the Quality of English LanguageNA
Round 2
Reviewer 2 Report
Comments and Suggestions for Authors
Dear Reviewer,
I appreciate changes and strongly recommend for acceptance.
Comments on the Quality of English Language
NA